# Combining Label Propagation and Simple Models out-performs Graph Neural Networks

**Qian Huang**[‡*]**, Horace He**[†*] **, Abhay Singh**[‡]**, Ser-Nam Lim**[§]**, Austin R. Benson**[‡]
Cornell University[‡], Facebook[†], Facebook AI[§]

## Abstract

Graph Neural Networks (GNNs) are a predominant technique for learning over graphs. However, there is relatively little understanding of why GNNs are successful in practice and whether they are necessary for good performance. Here, we show that for many standard transductive node classification benchmarks, we can exceed or match the performance of state-of-the-art GNNs by combining shallow models that ignore the graph structure with two simple post-processing steps that exploit correlation in the label structure: (i) an "error correlation" that spreads residual errors in training data to correct errors in test data and (ii) a "prediction correlation" that smooths the predictions on the test data. We call this overall procedure Correct and Smooth (C&S), and the post-processing steps are implemented via simple modifications to standard label propagation techniques that have long been used in graph-based semi-supervised learning. Our approach exceeds or nearly matches the performance of state-of-the-art GNNs on a wide variety of benchmarks, with just a small fraction of the parameters and orders of magnitude faster runtime. For instance, we exceed the best-known GNN performance on the OGB-Products dataset with 137 times fewer parameters and greater than 100 times less training time. The performance of our methods highlights how directly incorporating label information into the learning algorithm (as is common in traditional methods) yields easy and substantial performance gains. We can also incorporate our techniques into big GNN models, providing modest gains in some cases.

## 1 Introduction

Following the success of neural networks in computer vision and natural language processing, there are now a wide range of *graph neural networks* (GNNs) for making predictions involving relational data (Battaglia et al., 2018; Wu et al., 2020). These models have had much success and sit atop leaderboards such as the Open Graph Benchmark (Hu et al., 2020). Often, the methodological developments for GNNs revolve around creating strictly more expressive architectures than basic variants such as the Graph Convolutional Network (GCN) (Kipf & Welling, 2017) or GraphSAGE (Hamilton et al., 2017a); examples include Graph Attention Networks (Veličković et al., 2018), Graph Isomorphism Networks (Xu et al., 2018), and various deep models (Li et al., 2019; Rong et al., 2019; Chen et al., 2020). Many ideas for new GNN architectures are adapted from new architectures in models for language (e.g., attention) or vision (e.g., deep CNNs) with the hopes that success will translate to graphs. However, as these models become more complex, understanding their performance gains is a major challenge, and scaling them to large datasets is difficult.

Here, we see how far we can get by combining much simpler models, with an emphasis on understanding where there are easy opportunities for performance improvements in graph learning, particularly transductive node classification. We propose a simple pipeline with three main parts (Figure 1): (i) a base prediction made with node features that ignores the graph structure (e.g., a shallow multi-layer perceptron or just a linear model); (ii) a correction step, which propagates uncertainties from the training data across the graph to correct the base prediction; and (iii) a smoothing of the predictions over the graph. Steps (ii) and (iii) are post-processing and implemented with classical methods for graph-based semi-supervised learning, namely, label propagation techniques

---

[*]Equal contribution
[†]Work done while at Cornell University

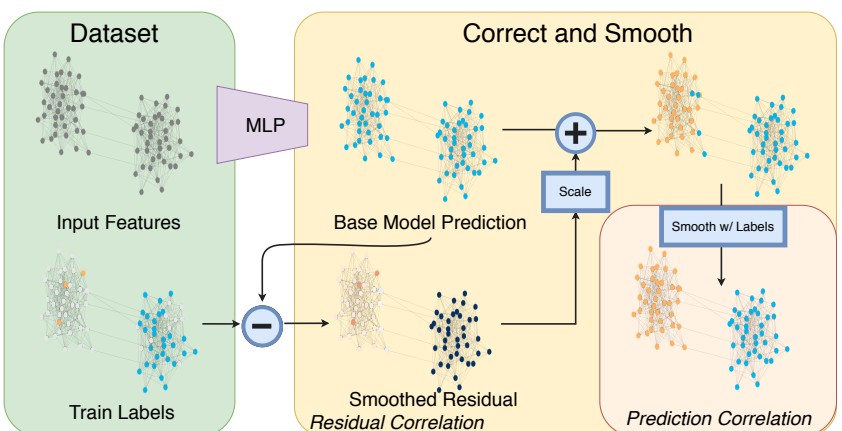

Figure 1: Illustration of our GNN-free model, Correct and Smooth (C&S), with a toy example. Nodes in the left and right clusters have different labels, marked by color (orange or blue). We use a multilayer perceptron (MLP) for base predictions, ignoring the graph structure. We assume this gives the same prediction on all nodes in this example (which could happen if, e.g., all nodes had the same features). After, base predictions are corrected by propagating errors from the training data. Finally, corrected predictions are smoothed with label propagation.

(Zhu, 2005).[1] With a few modifications and new deployment of these classic ideas, we achieve state-of-the-art performance on several node classification tasks, outperforming big GNN models. In our framework, the graph structure is not used to learn parameters (which is done in step (i)) but instead as a post-processing mechanism. This simplicity leads to models with orders of magnitude fewer parameters that take orders of magnitude less time to train and can easily scale to large graphs. We can also combine our ideas with state-of-the-art GNNs, although the performance gains are modest.

A major source of our performance improvements is directly using labels for predictions. This idea is not new — early diffusion-based semi-supervised learning algorithms on graphs such as the spectral graph transducer (Joachims, 2003), Gaussian random field models (Zhu et al., 2003), and and label spreading (Zhou et al., 2004) all use this idea. However, the motivation for these methods was semi-supervised learning on point cloud data, so the "node features" were used to construct the graph itself. Since then, these techniques have been used for learning on relational data consisting of a graph and some labels but no node features (Koutra et al., 2011; Gleich & Mahoney, 2015; Peel, 2017; Chin et al., 2019); however, they have largely been ignored in the context of GNNs. (That being said, we still find that even simple label propagation, which ignores features, does surprisingly well on a number of benchmarks.) This provides motivation for combining two orthogonal sources of prediction power — one coming from the node features (ignoring graph structure) and one coming from using the known labels directly in predictions.

Recent research connects GNNs to label propagation (Wang & Leskovec, 2020; Jia & Benson, 2020; 2021) as well as Markov Random fields (Qu et al., 2019; Gao et al., 2019), and some techniques use ad hoc incorporation of label information in the features (Shi et al., 2020). However, these approaches are usually still expensive to train, while we use label propagation in two understandable and low-cost ways. We start with a cheap "base prediction" from a model that uses only node features and ignores the graph structure. After, we use label propagation for error correction and then to smooth final predictions. These post-processing steps are based on the fact that errors and labels on connected nodes tend to be positively correlated. Assuming similarity between connected nodes is at the center of much network analysis and corresponds to homophily or assortative mixing (McPherson et al., 2001; Newman, 2003; Easley & Kleinberg, 2010). In the semi-supervised learning literature, the analog is the smoothness or cluster assumption (Chapelle et al., 2003; Zhu, 2005). The good performance of label propagation that we see across a wide variety of datasets suggests that these correlations hold on common benchmarks.

---

[1]One of the main methods that we use (Zhou et al., 2004) is often called *label spreading*. The term "label propagation" is used in a variety of contexts (Zhu, 2005; Wang & Zhang, 2007; Raghavan et al., 2007; Gleich & Mahoney, 2015). The salient point for this paper is that we assume positive correlations on neighboring nodes and that the algorithms work by "propagating" information from one node to another.

Overall, our methodology demonstrates that combining several simple ideas yields excellent performance in transductive node classification at a fraction of the cost, in terms of both model size (i.e., number of parameters) and training time. For example, on the OGB-Products benchmark, we out-perform the current best-known GNN with more than two orders of magnitude fewer parameters and more than two orders of magnitude less training time. However, our goal is *not* to say that current graph learning methods are poor or inappropriate. Instead, we aim to highlight easier ways in which to improve prediction performance in graph learning and to better understand the source of performance gains. Our main finding is that more direct incorporation of labels into the learning algorithms is key. We hope that our approach spurs new ideas that can help in other graph learning tasks, such as inductive node classification, link prediction, and graph prediction.

## 1.1 ADDITIONAL RELATED WORK

The Approximate Personalized Propagation of Neural Predictions (APPNP) framework is most relevant to our work, as they also smooth base predictions (Klicpera et al., 2018). However, they focus on integrating this smoothing into the training process so that their model can be trained end to end. Not only is this significantly more computationally expensive, it also prevents APPNP from incorporating label information at inference. Compared to APPNP, our framework produces more accurate predictions, is faster to train, and more easily scales to large datasets. That being said, APPNP can also be used without end-to-end training, which can make it faster but less accurate. Our framework also complements the Simplified Graph Convolution (Wu et al., 2019) and other algorithms designed to increase scalability (Bojchevski et al., 2020; Zeng et al., 2019; Frasca et al., 2020). The primary focus of our approach, however, is using labels directly, and scalability is a byproduct. There is also prior work connecting GCNs and label propagation. Wang & Leskovec (2020) use label propagation as a pre-processing step to weight edges for GNNs, whereas we use label propagation as a post-processing step and avoid GNNs. Jia & Benson (2020; 2021) use label propagation with GNNs for regression tasks, and our error correction step adapts some of their ideas for the case of classification. Finally, there are several recent approaches that incorporate nonlinearity into label propagation methods to compete with GNNs and achieve scalability (Eliav & Cohen, 2018; Ibrahim & Gleich, 2019; Tudisco et al., 2021), but these methods focus on settings of low label rates and do not use feature-based learning.

## 2 CORRECT AND SMOOTH (C&S) MODEL

We start with some notation. We assume that we have an undirected graph $G = (V, E)$, where there are $n = |V|$ nodes with features on each node represented by a matrix $X \in \mathbb{R}^{n \times p}$. Let $A$ be the adjacency matrix of the graph, $D$ be the diagonal degree matrix, and $S$ be the normalized adjacency matrix $D^{-1/2}AD^{-1/2}$. For the prediction problem, the node set $V$ is split into a disjoint set of unlabeled nodes $U$ and labeled nodes $L$, which are subsets of the indices $\{1, \ldots, n\}$. We will further split the labeled nodes into a training set $L_t$ and validation set $L_v$. We represent the labels by a one-hot-encoding matrix $Y \in \mathbb{R}^{n \times c}$, where $c$ is the number of classes (i.e., $Y_{ij} = 1$ if $i \in L$ is known to be in class $j$, and 0 otherwise, where the $i$th row of $Y$ is all zero if $i \in U$), Our problem is transductive node classification: assign each node $j \in U$ a label in $\{1, \ldots, c\}$, given $G$, $X$, and $Y$.

Our approach starts with a simple base predictor on node features that does not rely on any learning over the graph. After, we perform two types of label propagation (LP): one that corrects the base predictions by modeling correlated error and one that smooths the final prediction. We call the combination of these two methods Correct and Smooth (C&S; Figure 1). The LPs are only post-processing steps, and our pipeline is *not* trained end-to-end. Furthermore, the graph is only used in the post-processing steps (and in a pre-processing step to augment the features $X$), but not for the base predictions. This makes training fast and scalable compared to standard GNN models. Moreover, we take advantage of both LP (which performs fairly well on its own without features) and the node features. We find that combining these complementary signals yields excellent predictions.

## 2.1 SIMPLE BASE PREDICTOR

To start, we use a simple base predictor that does not rely on the graph structure. More specifically, we train a model $f$ to minimize $\sum_{i \in L_t} \ell(f(x_i), y_i)$, where $x_i$ is the $i$th row of $X$, $y_i$ is the $i$th row

of $Y$, and $\ell$ is a loss function, For this paper, $f$ is either a linear model or a shallow multi-layer perceptron (MLP) followed by a softmax, and $\ell$ is the cross-entropy loss. The validation set $L_v$ is used to tune hyperparameters such as learning rates and the hidden layer dimensions for the MLP. From $f$, we get a *base prediction* $Z \in \mathbb{R}^{n \times c}$, where each row of $Z$ is a probability distribution resulting from the softmax. Omitting the graph structure for these base predictions avoids most of the scalability issues with GNNs. In principle, though, we can use any base predictor for $Z$, including those based on GNNs, and we explore this in Section 3. However, for our pipeline to be simple and scalable, we just use linear classifiers or MLPs with subsequent post-processing, which we describe next.

## 2.2 Correcting base predictions with error correlation

Next, we improve the accuracy of the base prediction $Z$ by incorporating labels to correlate errors. The key idea is that we expect *errors* in the base prediction to be positively correlated along edges in the graph. In other words, an error at node $i$ increases the chance of a similar error at neighbors of $i$. Thus, we should "spread" such uncertainty over the graph. Our approach here is inspired in part by residual propagation (Jia & Benson, 2020), where a similar concept is used for node regression tasks, as well as generalized least squares and correlated error models more broadly (Shalizi, 2013). To this end, we first define an error matrix $E \in \mathbb{R}^{n \times c}$, where error is the *residual* on the training data and zero elsewhere:

$$E_{L_t,:} = Y_{L_t,:} - Z_{L_t,:}, \quad E_{L_v,:} = 0, \quad E_{U,:} = 0. \tag{1}$$

The residuals in rows of $E$ corresponding to training nodes are zero only when the base predictor makes a perfect prediction. We smooth the error using the label spreading technique of Zhou et al. (2004), optimizing the objective

$$\hat{E} = \underset{W \in \mathbb{R}^{n \times c}}{\arg\min} \ \text{trace}(W^T(I - S)W) + \mu \|W - E\|_F^2. \tag{2}$$

The first term encourages smoothness of the error estimation over the graph, and is equal to $\sum_{k=1}^{c} \sum_{(i,j) \in E} (W_{ik}/\sqrt{D_{ii}} - W_{jk}/\sqrt{D_{jj}})^2$. The second term keeps the solution close to the initial guess $E$ of the error. As derived in Zhou et al. (2004), the solution can be obtained via the iteration $E^{(t+1)} = (1 - \alpha)E + \alpha S E^{(t)}$, where $\alpha = 1/(1 + \mu)$ and $E^{(0)} = E$, which converges rapidly to $\hat{E}$. This iteration is a propagation (or diffusion or spreading) of the error, and we add the smoothed errors to the base prediction to get corrected predictions $Z^{(r)} = Z + \hat{E}$. We emphasize that this is a post-processing technique and there is no coupled training with the base predictions.

This type of propagation is motivated by a particular correlated Gaussian error assumption for regression problems (Jia & Benson, 2020; 2021). For the classification problems we consider, we find that the smoothed errors $\hat{E}$ might not be at the right scale. We know that

$$\|E^{(t+1)}\|_2 \leq (1 - \alpha)\|E\| + \alpha\|S\|_2\|E^{(t)}\|_2 = (1 - \alpha)\|E\|_2 + \alpha\|E^{(t)}\|_2. \tag{3}$$

When $E^{(0)} = E$, we then have that $\|E^{(t)}\|_2 \leq \|E\|_2$. Thus, the propagation cannot completely correct the errors on all nodes in the graph, as it does not have enough "total mass," and we find that adjusting the scale of the residual can help substantially in practice. To do this, we propose two variations of scaling the residual.

**Autoscale.** Intuitively, we want to scale the size of errors in $\hat{E}$ to be approximately the size of the errors in $E$. We only know the true errors at labeled nodes, so we approximate the scale with the average error over the training nodes. Formally, let $e_j^T \in \mathbb{R}^c$ and $\hat{e}_j^T$ correspond to the $j$th rows of $E$ and $\hat{E}$ and define $\sigma = \frac{1}{|L_t|} \sum_{j \in L_t} \|e_j\|_1$. Then we define the corrected predictions on an unlabeled node $i \in U$ to be $Z_{i,:}^{(r)} = Z_{i,:} + \sigma/\|\hat{e}_i\|_1 \cdot \hat{e}_i^T$.

**Scaled Fixed Diffusion (FDiff-scale).** Alternatively, we can use a diffusion like the one from Zhu et al. (2003), which keeps the known errors at training nodes *fixed*. More specifically, we iterate $E_{U,:}^{(t+1)} = [D^{-1}AE^{(t)}]_{U,:}$ and keep fixed $E_{L,:}^{(t)} = E_{L,:}$ until convergence to $\hat{E}$, starting with $E^{(0)} = E$. Intuitively, this fixes error values where we know the error (on the labeled nodes $L$), while other nodes keep averaging over the values of their neighbors until convergence. With this type of propagation, the maximum and minimum values of entries in $E^{(t)}$ do not go beyond those in $E_L$. We still find it effective to select a scaling hyperparameter $s$ to produce $Z^{(r)} = Z + s\hat{E}$.

## 2.3 Smoothing final predictions with Prediction Correlation

At this point, we have a score vector $Z^{(r)}$, obtained from correcting the base predictor $Z$ with a model for the correlated error $\hat{E}$. To make a final prediction, we further smooth the corrected predictions. The motivation is that adjacent nodes in the graph are likely to have similar labels, which is expected given network homophily or assortative properties of a network. Thus, we can encourage smoothness over the distribution over labels by another label propagation. First, we start with our best guess $H \in \mathbb{R}^{n \times c}$ of the labels:

$$H_{L_t,:} = Y_{L_t,:}, \quad H_{L_v \cup U,:} = Z^{(r)}_{L_v \cup U,:}. \tag{4}$$

Here, the true labels are used at the training nodes and the corrected predictions are used for the validation and unlabeled nodes, the latter of which no longer correspond to probability distributions. We can (and should) also use the true labels at the validation labels, which we discuss later in the experiments, but the setup in Equation (4) aligns more closely with standard GNN evaluation. We then iterate $H^{(t+1)} = (1 - \alpha)H + \alpha S H^{(t)}$ with $H^{(0)} = H$ until convergence to give the final prediction $\hat{Y}$. The classification for a node $i \in U$ is $\arg\max_{j \in \{1,...,c\}} \hat{Y}_{ij}$.

As with error correlation, the smoothing here is a post-processing step, decoupled from the other steps. This type of prediction smoothing is similar in spirit to APPNP (Klicpera et al., 2018), which we compare against later. However, APPNP is typically trained end-to-end, propagates final-layer representations instead of softmaxes, does not use labels, and is motivated differently.

## 2.4 Summary and additional considerations

To summarize, we start with a cheap base prediction $Z$, using only node features but not the graph structure. After, we estimate errors $\hat{E}$ by propagating errors on the training data. Then, we add these errors back to the base predictions, forming corrected predictions. Finally, we treat the corrected predictions as score vectors on unlabeled nodes, and combine them with the known labels via another LP step for smoothed final predictions. We call this pipeline **Correct and Smooth** (C&S).

Before showing that this pipeline achieves state-of-the-art performance on transductive node classification, we briefly describe another simple way of improving performance: feature augmentation. The hallmark of deep learning is that we can learn features instead of engineering them. However, GNNs still rely on informative input features to make predictions. There are numerous ways to get useful features from just the graph topology to augment the raw node features (Henderson et al., 2011; 2012; Hamilton et al., 2017b). In our pipeline, we augment features with a regularized spectral embedding (Chaudhuri et al., 2012; Zhang & Rohe, 2018) coming from the leading $k$ eigenvectors of the matrix $D_\tau^{-1/2}(A + \frac{\tau}{n}\mathbf{1}\mathbf{1}^T)D_\tau^{-1/2}$, where $\mathbf{1}$ is a vector of all ones, $\tau$ is a regularization parameter set to the average degree, and $D_\tau$ is diagonal with $i$th diagonal entry equal to $D_{ii} + \tau$. The underlying matrix is dense, but we can apply matrix-vector products in time linear in the number of edges and use iterative eigensolvers to compute the embeddings quickly.

## 3 Experiments on Transductive Node Classification

To demonstrate the effectiveness of our methods, we use nine datasets (Table 1). The Arxiv and Products datasets are from the Open Graph Benchmark (OGB) (Hu et al., 2020); the Cora, Citeseer, and Pubmed are three classic citation network benchmarks (Getoor et al., 2001; Getoor, 2005; Namata et al., 2012); and wikiCS is a web graph (Mernyei & Cangea, 2020). In these datasets, classes are categories of papers, products, or pages, and features are derived from text. We also use a Facebook social network of Rice University, where classes are dorm residences and features are attributes such as gender, major, and class year (Traud et al., 2012), as well as a geographic dataset of US counties where classes are 2016 election outcomes and features are demographic (Jia & Benson, 2020). Finally, we use an email dataset of a European research institute, where classes are department membership and there are no features (Leskovec et al., 2007; Yin et al., 2017).

**Data splits.** The training/validation/test splits for Arxiv and Products are given by the benchmark, and the splits for wikiCS come from Mernyei & Cangea (2020). For the Rice, US counties, and email data, we use 40%/10%/50% random splits. For the smaller citation networks, we use 60%/20%/20%

Table 1: Summary statistics of datasets and model performance. For the accuracy of our best C&S model compared to the state-of-the-art GNN method (see text), we report the change in the number of parameters and the accuracy. We also list the training time with time to compute the spectral embedding in parentheses (even if not used in the best model). Our methods require fewer parameters, are typically more accurate, and are fast to train. Also see Tables 2 and 3.

| Datasets | Classes | Nodes | Edges | Parameter $\Delta$ | Accuracy $\Delta$ | Time (s) |
|----------|---------|-------|-------|-------------------|-------------------|----------|
| Arxiv | 40 | 169,343 | 1,166,243 | −84.90% | +0.26 | 12 (+90) |
| Products | 47 | 2,449,029 | 61,859,140 | −93.47% | +1.74 | 171 (+2959) |
| Cora | 7 | 2,708 | 5,429 | −98.37% | +1.09 | < 1 (+7) |
| Citeseer | 6 | 3,327 | 4,732 | −89.68% | −0.69 | < 1 (+7) |
| Pubmed | 3 | 19,717 | 44,338 | −96.00% | −0.30 | < 1 (+14) |
| Email | 42 | 1,005 | 25,571 | −97.89% | +4.33 | 43 (+17) |
| Rice31 | 10 | 4,087 | 184,828 | −99.02% | +1.39 | 39 (+12) |
| US County | 2 | 3,234 | 12,717 | −74.56% | +1.77 | 39 (+12) |
| wikiCS | 10 | 11,701 | 216,123 | −84.88% | +2.03 | 7 (+11) |

random splits, as in Wang & Leskovec (2020). Standard deviations in prediction accuracy over splits is < 1% in most experiments and such variance does not change our qualitative comparisons.

**C&S setup and baselines.** We use *Linear* and *MLP* models as simple base predictors based on node features. When a spectral embedding is included as a node feature, we refer to these models as *Linear-SE* and *MLP-SE*. We also evaluate Label Propagation itself (*LP*; specifically, the Zhou et al. (2004) version), which only uses labels. In all cases, the number of LP iterations is fixed to 50.

For GNN models comparable to our framework in terms of simplicity or style, we use GCN, SGC, and APPNP. For GCNs, we add residual connections from the input to every layer and from every layer to the output, as well as dropout. Thus, *GCN* is not the original model Kipf & Welling (2017) and instead serves as a fairly strong representative of out-of-the-box GNN capabilities. The number of layers and hidden layer dimensions for the GCNs are the same as the MLPs used by our base predictors. The GCN only uses raw node features, and additional results in Appendix C show that including spectral embeddings minimally changes performance. APPNP uses a linear model for base predictions, also with the raw node features.

Finally, we include several "state-of-the-art" (SOTA) baselines. For Arxiv and Products, this is UniMP (Shi et al., 2020) (top of OGB leaderboard, as of October 1, 2020). For Cora, Citeseer and Pubmed, we use the top scores from Chen et al. (2020). For Email and US County, we use GCNII (Chen et al., 2020). For Rice31, we use GCN with spectral embedding as additional features, which is the best GNN-based model that we found. For wikiCS, we use APPNP as reported by Mernyei & Cangea (2020). Hyperparameters are tuned using the validation set.

All of the above models select hyperparameters using the validation set. See Appendix A for additional model architecture details.

## 3.1 FIRST RESULTS ON NODE CLASSIFICATION

In our first set of results, we only use the training labels in our C&S framework, as these are what GNNs typically use to train models. For the results discussed here, this is generous to our baselines. The ability to include validation labels is an advantage of our approach (and LP in general), and this improves performance of our framework even further (Table 1). We discuss this in the next section.

Table 2 reports the results, and we highlight a few important findings. First, within our model, there are substantial gains from the LP post-processing steps (e.g., the MLP-SE base prediction accuracy increases from 63% to 84% on Products). Second, even Linear with C&S outperforms GCNs in many cases, and simple LP is often competitive with GCNs. This is striking given that the main motivation for GCNs was to address the fact that connected nodes may not have similar labels (Kipf & Welling, 2017). Our results suggest that directly incorporating correlation in the graph with simple use of the features is often a better idea. Results in Appendix B show that both label propagation post-processing steps are important for performance. Third, our model variants can out-perform SOTA on Products, Cora, Email, Rice31, and US County (often substantially so). On the other datasets, there is not much difference between the best C&S model and the SOTA.

Table 2: Performance of our C&S framework, using only the training labels as ground truth in final prediction smoothing (Equation (4)). Further improvements can be made by including ground truth validation labels (Table 3). The Email dataset has no raw node features, so some methods are not evaluated. APPNP ran out of memory (OOM) on the products dataset.

| Method | Arxiv | Products | Cora | Citeseer | Pubmed |
|---|---|---|---|---|---|
| LP | 68.5 | 74.76 | 86.50 | 70.64 | 83.74 |
| GCN | 71.74 | 75.64 | 85.77 | 73.68 | 88.13 |
| SGC | 69.39 | 68.83 | 86.81 | 72.04 | 84.04 |
| APPNP | 66.38 | OOM | 87.87 | 76.53 | 89.40 |
| SOTA | **73.79** | 82.56 | 88.49 | **77.99** | **90.30** |
| Linear | 52.32 | 47.73 | 73.85 | 70.27 | 87.10 |
| Linear-SE | 70.08 | 50.05 | 74.75 | 70.51 | 87.19 |
| MLP-SE | 71.51 | 63.41 | 74.06 | 68.10 | 86.85 |
| Linear + C&S (Autoscale) | 71.11 | 80.24 | 88.62 | 76.31 | 89.99 |
| Linear-SE + C&S (Autoscale) | 72.07 | 80.25 | 88.73 | 76.75 | 89.93 |
| MLP-SE + C&S (Autoscale) | 72.62 | 78.60 | 87.39 | 76.31 | 89.33 |
| Linear + C&S (Fdiff-scale) | 70.60 | 82.54 | **89.05** | 76.22 | 89.74 |
| Linear-SE + C&S (Fdiff-scale) | 71.57 | 83.01 | 88.66 | 77.06 | 89.51 |
| MLP-SE + C&S (Fdiff-scale) | 72.43 | **84.18** | 87.39 | 76.42 | 89.23 |
| Method | Email | Rice31 | US County | wikiCS | |
| LP | 70.69 | 82.19 | 87.90 | 76.72 | |
| GCN | — | 15.45 | 84.13 | 78.61 | |
| SGC | — | 16.59 | 83.92 | 72.86 | |
| APPNP | — | 11.34 | 84.14 | 69.83 | |
| SOTA | 71.96 | 86.50 | 88.08 | **79.84** | |
| Linear | — | 9.84 | 75.74 | 72.45 | |
| Linear-SE | 66.24 | 70.26 | 84.07 | 74.29 | |
| MLP-SE | 69.13 | 17.16 | 87.70 | 73.07 | |
| Linear + C&S (Autoscale) | — | 75.99 | 85.25 | 79.57 | |
| Linear-SE + C&S (Autoscale) | 72.50 | 86.42 | 86.15 | 79.53 | |
| MLP-SE + C&S (Autoscale) | 74.55 | 85.50 | 89.64 | 78.10 | |
| Linear + C&S (Fdiff-scale) | — | 73.66 | 87.38 | 79.54 | |
| Linear-SE + C&S (Fdiff-scale) | 72.53 | **87.55** | 88.11 | 79.25 | |
| MLP-SE + C&S (Fdiff-scale) | **75.74** | 85.74 | **89.85** | 78.24 | |

To get a sense of how much using ground truth labels directly helps, we also evaluate a version of C&S where we smooth base predictions from a linear model or MLP, using the Zhou et al. (2004) version of label propagation. We call these *Linear-SE-smooth* and *MLP-SE-smooth* and find that they often outperform GCNs (right). Again,

| Method | Arxiv | Products |
|---|---|---|
| Linear-SE-smooth | 71.42 | 78.73 |
| MLP-SE-smooth | **72.48** | **80.34** |
| GCN | 71.74 | 75.64 |

these results suggest that smoothed outputs are important, aligning with recent research (Wu et al., 2019; Bojchevski et al., 2020), and that the original motivations for GCNs might be misleading. However, there are still gaps in performance between these models and those in Table 2 that directly use labels. Next, we see how to improve performance of C&S even further by using more labels.

## 3.2 FURTHER IMPROVEMENTS BY USING MORE LABELS

We improve the C&S performance by using both training and validation labels, instead of just the training labels as in Equation (4). Importantly, we do *not* use validation labels to update the base prediction model — they are just used to select hyperparameters. Using validation labels boosts performance even further: Table 3 shows accuracies and Table 1 shows gains over SOTA. The ability to incorporate validation labels is a benefit of our approach. On the other hand, GNNs do not have this advantage, as they often rely on early stopping to prevent overfitting, may not always

Table 3: Performance of C&S, using both training and validation labels as ground truth in the final prediction smoothing (cf. Equation (4), Table 2).

| Method | Arxiv | Products | Cora | Citeseer | Pubmed |
|---|---|---|---|---|---|
| Linear + C&S (Autoscale) | 72.71 | 80.55 | 89.54 | 76.83 | 90.01 |
| Linear-SE + C&S (Autoscale) | 73.78 | 80.56 | **89.77** | 77.11 | 89.98 |
| MLP-SE + C&S (Autoscale) | **74.02** | 79.29 | 88.55 | 76.36 | 89.50 |
| Linear + C&S (Fdiff-scale) | 72.42 | 82.89 | 89.47 | 77.08 | 89.74 |
| Linear-SE + C&S (Fdiff-scale) | 72.93 | 83.27 | 89.53 | 77.29 | 89.57 |
| MLP-SE + C&S (Fdiff-scale) | 73.46 | **84.55** | 88.18 | 76.41 | 89.38 |
| SOTA | 73.65 | 82.56 | 88.49 | **77.99** | **90.30** |

| Methods | Email | Rice31 | US County | wikiCS | |
|---|---|---|---|---|---|
| Linear + C&S (Autoscale) | — | 76.59 | 85.22 | **81.87** | |
| Linear-SE + C&S (Autoscale) | 73.33 | 87.25 | 86.38 | 81.57 | |
| MLP-SE + C&S (Autoscale) | 73.45 | 86.13 | 89.71 | 80.75 | |
| Linear + C&S (Fdiff-scale) | — | 75.31 | 88.16 | 81.18 | |
| Linear-SE + C&S (Fdiff-scale) | 72.57 | **87.89** | 88.06 | 81.06 | |
| MLP-SE + C&S (Fdiff-scale) | **76.22** | 86.26 | **90.05** | 80.83 | |
| SOTA | 71.96 | 86.50 | 88.08 | 79.84 | |

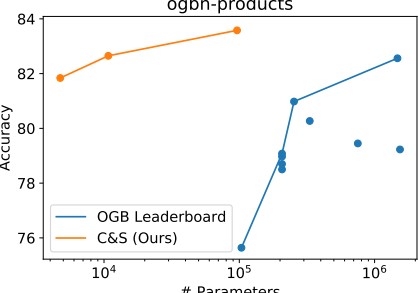

Figure 2: Accuracy and model size on Products.

Table 4: C&S with GNN base predictions.

| Dataset | Model | Performance |
|---|---|---|
| ogbn-arxiv | GAT | 73.56 |
| | GAT + C&S | **73.86** |
| | SOTA | 73.79 |
| US County | GCNII (SOTA) | 88.08 |
| | GCNII + C&S | **89.59** |

benefit from more data (e.g., under distributional shift), and do not directly use labels. Thus, our comparisons in Table 2 are more generous than needed. With validation labels, our best model out-performs SOTA in seven of nine datasets, often by substantial margins (Table 1).

The evaluation procedure for GNN benchmarks differ from those for LP. For GNNs, a sizable validation set is often used (and needed) for substantial hyperparameter tuning, as well as early stopping. With LP, one can use the entire set of labeled nodes $L$ with cross-validation to select the single hyperparameter $\alpha$. Given the setup of transductive node classification, there is no reason not to use validation labels at inference if they are helpful (e.g., via LP in our case). The results in Tables 1 and 3 show the true performance of our model and is the proper point of comparison.

Overall, our results highlight two important findings. First, big and expensive-to-train GNN models are not actually necessary to achieve top performance for transductive node classification on many datasets. Second, combining classical label propagation ideas with simple base predictors outperforms graph neural networks on these tasks.

### 3.3 TRAINING TIME AND IMPROVING EXISTING GNNS

Our C&S framework often has significantly fewer parameters compared to GNNs or other SOTA solutions. As an example, we plot parameters vs. performance for the Products dataset in Figure 2. While having fewer parameters is useful, the real gain is in faster training time. Our models are typically orders of magnitude faster to train than models with comparable accuracy because we do not use the graph structure for our base prediction models. As one example, although our MLP-SE + C&S model for the Arxiv dataset has a similar number of parameters compared to the

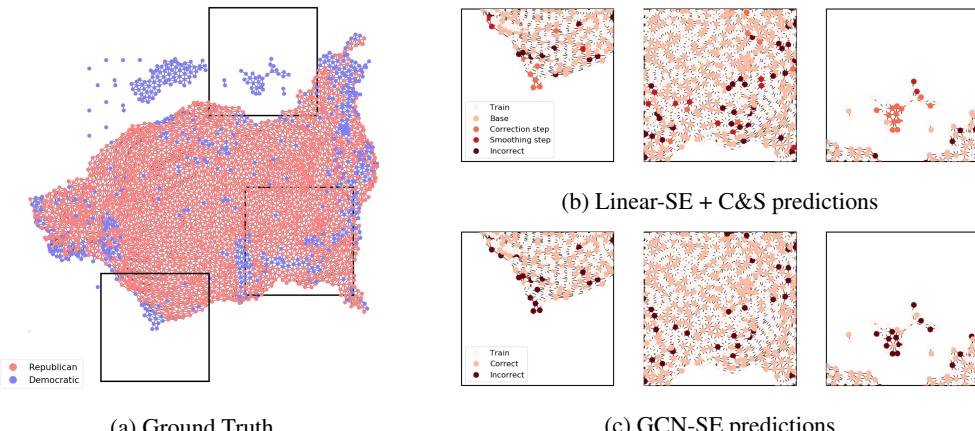

(a) Ground Truth

(b) Linear-SE + C&S predictions

(c) GCN-SE predictions

Figure 3: (a) US County visualizations, where the embedding is given by GraphViz and colors correspond to class labels. (b) Panels corresponding to parts of (a) that show at which stage Linear-SE + C&S made a correct prediction. (c) The same panels showing GCN predictions.

"GCN+linear+labels" method on the OGB leaderboard (Wang, 2020), our model runs 7 times faster per epoch and converges much faster. In addition, compared to the SOTA for the Products dataset, *our framework with a linear base predictor has higher accuracy, trains over 100 times faster, and has 137 times fewer parameters*.

We also evaluated our methods on an even larger dataset, the papers100M OGB benchmark (Hu et al., 2020). Here, we obtain 65.33% using C&S with the Linear model as the base predictor, which out-performs the state-of-the-art (63.29%, as of October 1, 2020).

Our pipeline can also be used to improve the performance of GNNs in general. We used C&S with base predictions given by GCNII or GAT. This improves our results on some datasets, such as ogbn-arxiv (Table 4). However, the performance improvements are sometimes only minor, suggesting that big models might be capturing the same signal as our simple C&S framework.

### 3.4 PERFORMANCE VISUALIZATION

To aid in understanding the performance of our C&S framework, we visualize the predictions on the US County dataset (Figure 3). As expected, the residual error correlation tends to correct nodes where neighboring counties provide relevant information. For example, we see that many errors in the base predictions are corrected by the residual correlation (Figure 3b, left and right panels). In these cases, which correspond to parts of Texas and Hawaii, the demographic features of the counties are outliers compared to the rest of the country, leading both the linear model and GCN astray. The error correlation from neighboring counties is able to fix the predictions. We also see that the final prediction correlation can fix errors when nearby nodes are correctly classified, as shown in the center panel of Figure 3b. We observe similar behavior on the Rice31 dataset (Appendix D).

## 4 DISCUSSION

GNN models are becoming more expressive, more parameterized, and more expensive to train. Our results suggest that we should explore other techniques for improving performance, such as label propagation and feature augmentation. In particular, label propagation and its variants are longstanding, powerful ideas. More directly incorporating them into graph learning models has major benefits, and we have shown that these can lead to both better predictions and faster training.

**Acknowledgments.** This research was supported by Facebook AI, NSF Award DMS-1830274, ARO Award W911NF19-1-0057, ARO MURI, and JP Morgan Chase & Co. We also thank Cornell University Artificial Intelligence for their support, as well as Marc Brockschmidt, Matthias Fey, Stephan Günnemann, Weihua Hu, and Junteng Jia for insightful discussions.

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

## A    MODEL DETAILS

Here we provide some more details on the models that we use. In all cases we use the Adam optimizer and tune the learning rate. We follow the models and hyperparameters provided in OGB (Hu et al., 2020) and wikiCS (Mernyei & Cangea, 2020) and manually tune some hyperparameters on the validation data for the potential of better performance.

For our MLPs, every linear layer is followed by batch normalization, ReLU activation, and 0.5 dropout. The other parameters depend on the dataset as follows.

- Products and Arxiv: 3 layers and 256 hidden channels with learning rate equal to 0.01.
- Cora, Citseer, and Pubmed (Getoor et al., 2001; Getoor, 2005; Namata et al., 2012) and Email (Leskovec et al., 2007; Yin et al., 2017): 3 layers and 64 hidden channels with learning rate = 0.01.
- wikiCS: 3 layers and 256 hidden channels with learning rate equal to 0.005.
- US County (Jia & Benson, 2020) and Rice31 (Traud et al., 2012): 5 layers and 256 hidden channels with learning rate equal to 0.005.

SOTA models for most datasets are taken from existing benchmarks. We determined SOTA for Email, US County, and Rice31 by evaluating several models discussed in the paper. The best performing baselines were as follows. For Email, GCNII with 5 layers, 256 hidden channels, learning rate equal to 0.01. For US County, GCNII with 8 layers, 256 hidden channels, learning rate equal to 0.03. For Rice31, we reused our GCN architecture and trained it over spectral embedding, which substantially outperformed the other GNN variants.

All models were implemented with PyTorch (Paszke et al., 2019) and PyTorch Geometric (Fey & Lenssen, 2019).

## B    PERFORMANCE RESULTS WITH ONLY THE CORRECTION STEP

Table 5 shows results with and without smoothing in the final predictions, i.e., just the "C step" vs. C&S. Including final prediction smoothing provides a substantial performance boost in many cases.

## C    ANALYSIS OF PERFORMANCE GAINS FROM SPECTRAL EMBEDDINGS

Table 6 shows the effect of including spectral embeddings as node features on the accuracy of the MLP-based and GCN models. In the case of the Arxiv dataset, including the spectral embedding improves the MLP base prediction performance substantially and the C&S performance modestly, but hardly changes the performance of the GCN. For Pubmed, including the spectral embeddings barely changes the performance of any model.

## D    ADDITIONAL VISUALIZATION

Full visualizations of C&S and GCN-SE performance for the US County dataset are in Figures 4 to 6. Similar visualizations for the Rice31 are in Figures 7 to 9, which are generated by projecting the 128-dimensional spectral embedding used in the main text down to two dimensions with UMAP (McInnes et al., 2018).

Table 5: Performance of our C&S framework with and without the final prediction smoothing. In cases where final prediction smoothing is used, only ground truth training are used.

| Method | Arxiv | Products | Cora | Citeseer | Pubmed |
|---|---|---|---|---|---|
| Linear + C (Autoscale) | 66.89 | 74.63 | 79.56 | 72.56 | 88.56 |
| Linear + C&S (Autoscale) | 71.11 | 80.24 | 88.62 | 76.31 | 89.99 |
| Linear-SE + C (Autoscale) | 71.52 | 70.93 | 79.08 | 70.77 | 88.84 |
| Linear-SE + C&S (Autoscale) | 72.07 | 80.25 | 88.73 | 76.75 | 89.93 |
| MLP-SE + C (Autoscale) | 71.97 | 69.85 | 74.11 | 71.78 | 87.35 |
| MLP-SE + C&S (Autoscale) | 72.62 | 78.60 | 87.39 | 76.31 | 89.33 |
| Linear + C (Fdiff-scale) | 65.62 | 80.97 | 76.48 | 70.48 | 87.52 |
| Linear + C&S (Fdiff-scale) | 70.60 | 82.54 | 89.05 | 76.22 | 89.74 |
| Linear-SE + C (Fdiff-scale) | 70.26 | 73.89 | 79.32 | 70.53 | 84.47 |
| Linear-SE + C&S (Fdiff-scale) | 71.57 | 83.01 | 88.66 | 77.06 | 89.51 |
| MLP-SE + C (Fdiff-scale) | 71.55 | 72.72 | 74.36 | 71.45 | 86.97 |
| MLP-SE + C&S (Fdiff-scale) | 72.43 | 84.18 | 87.39 | 76.42 | 89.23 |
| Method | Email | Rice31 | US County | wikiCS | |
| Linear + C (Autoscale) | — | 43.97 | 82.60 | 77.49 | |
| Linear + C&S (Autoscale) | — | 75.99 | 85.25 | 79.57 | |
| Linear-SE + C (Autoscale) | 73.39 | 86.19 | 84.08 | 74.06 | |
| Linear-SE + C&S (Autoscale) | 72.50 | 86.42 | 86.15 | 79.53 | |
| MLP-SE + C (Autoscale) | 71.64 | 84.61 | 88.83 | 78.72 | |
| MLP-SE + C&S (Autoscale) | 74.55 | 85.50 | 89.64 | 78.10 | |
| Linear + C (Fdiff-scale) | — | 72.44 | 87.16 | 75.98 | |
| Linear + C&S (Fdiff-scale) | — | 73.66 | 87.38 | 79.54 | |
| Linear-SE + C (Fdiff-scale) | 71.31 | 85.22 | 88.27 | 73.86 | |
| Linear-SE + C&S (Fdiff-scale) | 72.53 | 87.55 | 88.11 | 79.25 | |
| MLP-SE + C (Fdiff-scale) | 72.59 | 85.42 | 89.62 | 78.40 | |
| MLP-SE + C&S (Fdiff-scale) | 75.74 | 85.74 | 89.85 | 78.24 | |

Table 6: Comparison of models with and without spectral embeddings, using only ground truth training labels for final prediction smoothing within C&S.

| Method | Arxiv | Products | Cora | Citeseer | Pubmed |
|---|---|---|---|---|---|
| GCN | 71.74 | 75.64 | 85.77 | 73.68 | 88.13 |
| GCN-SE | 71.76 | 76.12 | 85.83 | 73.60 | 88.32 |
| MLP | 59.67 | 59.23 | 74.21 | 69.34 | 86.73 |
| MLP-SE | 71.51 | 63.41 | 74.06 | 68.10 | 86.85 |
| MLP + C&S (Autoscale) | 71.76 | 79.42 | 87.56 | 76.42 | 89.29 |
| MLP-SE + C&S (Autoscale) | 72.62 | 78.60 | 87.39 | 76.31 | 89.33 |
| MLP + C&S (FDiff-scale) | 71.57 | 83.8 | 87.61 | 76.44 | 89.28 |
| MLP-SE + C&S (Fdiff-scale) | 72.43 | 84.18 | 87.39 | 76.42 | 89.23 |

| Method | Email | Rice31 | US County | wikiCS |
|---|---|---|---|---|
| GCN | — | 15.45 | 84.13 | 78.61 |
| GCN-SE | 74.51 | 38.54 | 89.72 | 78.15 |
| MLP | — | 15.73 | 87.77 | 71.42 |
| MLP-SE | 69.13 | 17.16 | 87.70 | 73.07 |
| MLP + C&S (Autoscale) | — | 85.05 | 89.67 | 78.92 |
| MLP-SE + C&S (Autoscale) | 74.55 | 85.50 | 89.64 | 78.10 |
| MLP + C&S (FDiff-scale) | — | 86.40 | 89.64 | 78.10 |
| MLP-SE + C&S (Fdiff-scale) | 75.74 | 85.74 | 89.85 | 78.24 |

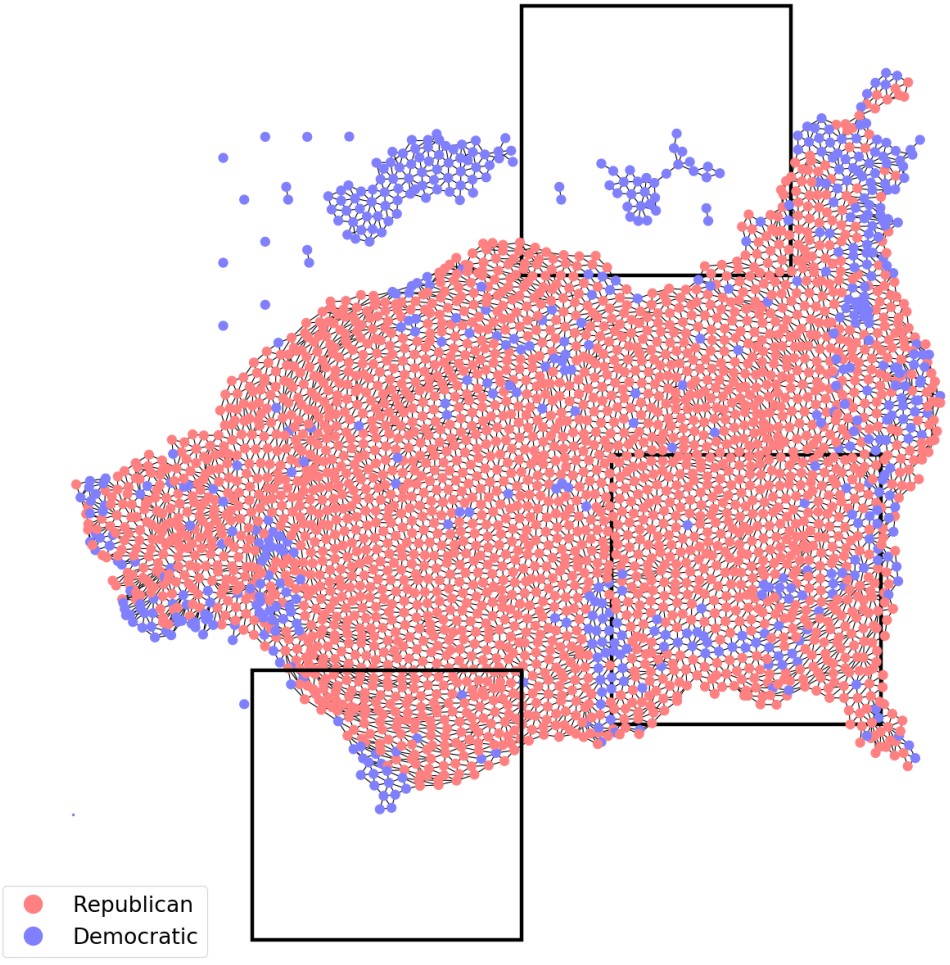

Figure 4: US County ground truth class labels.

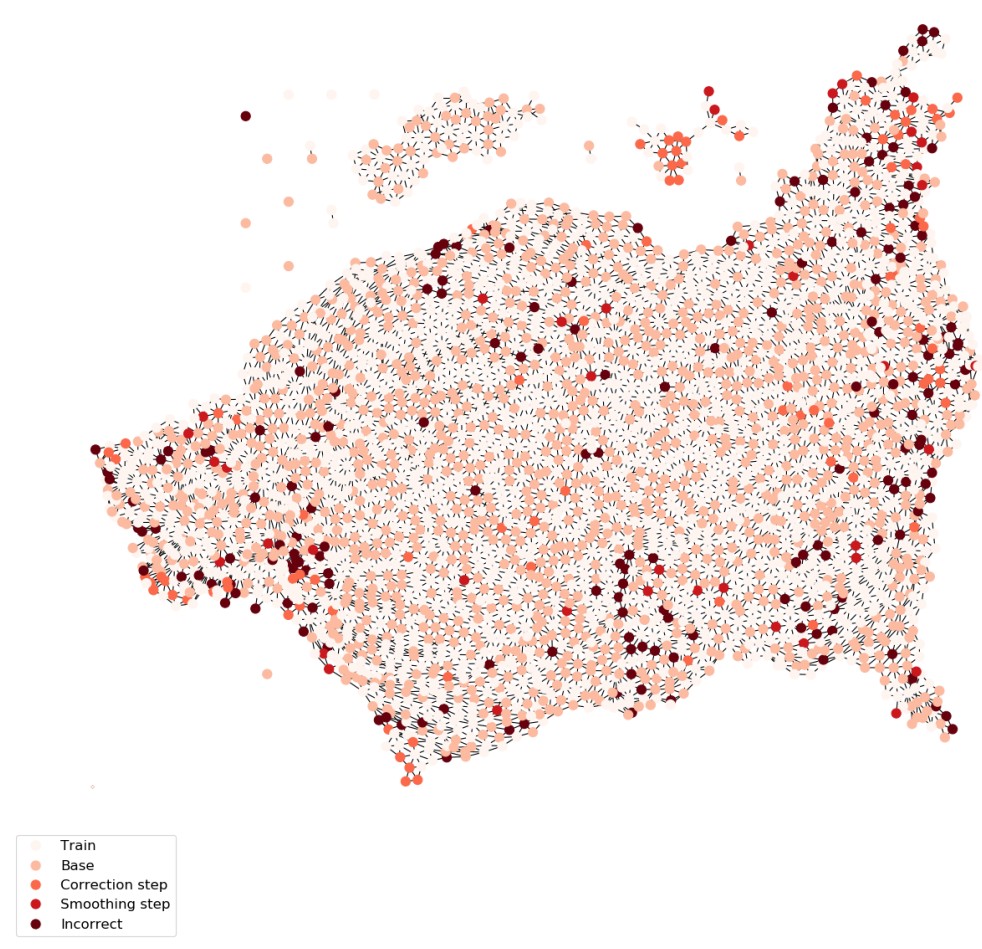

Figure 5: Linear-SE + C&S prediction performance on US County.

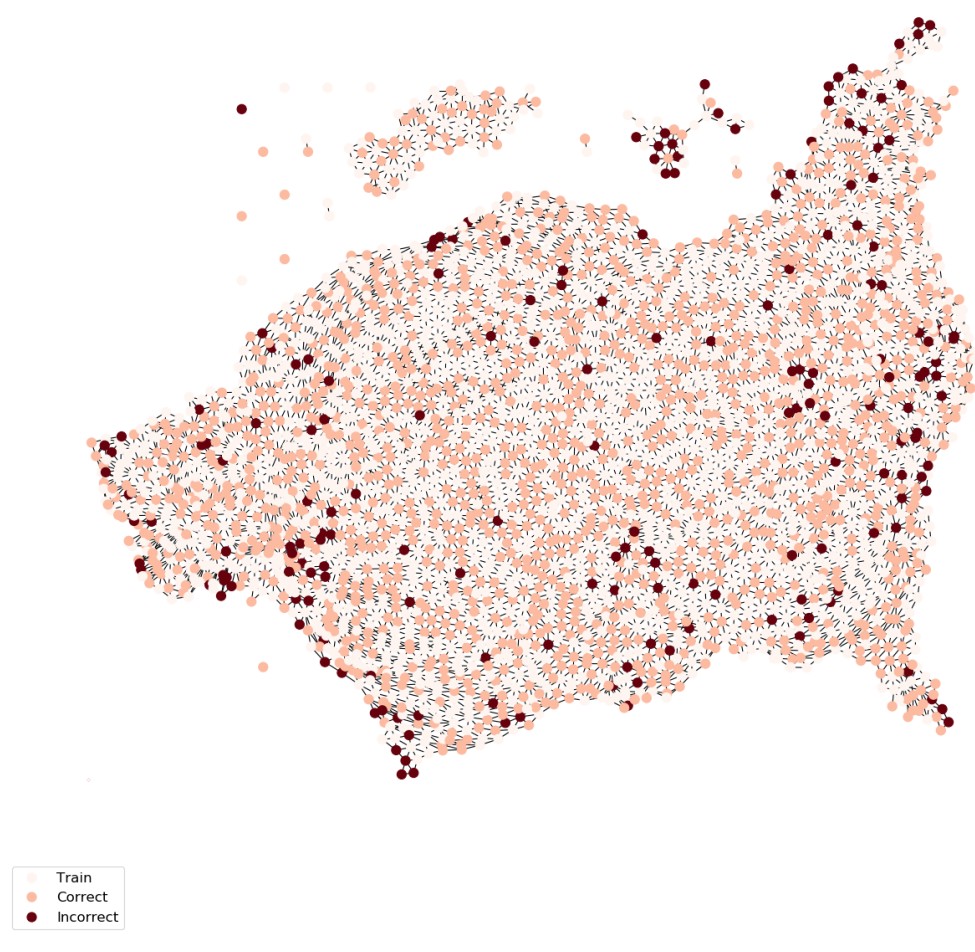

Figure 6: GCN-SE prediction performance on US County.

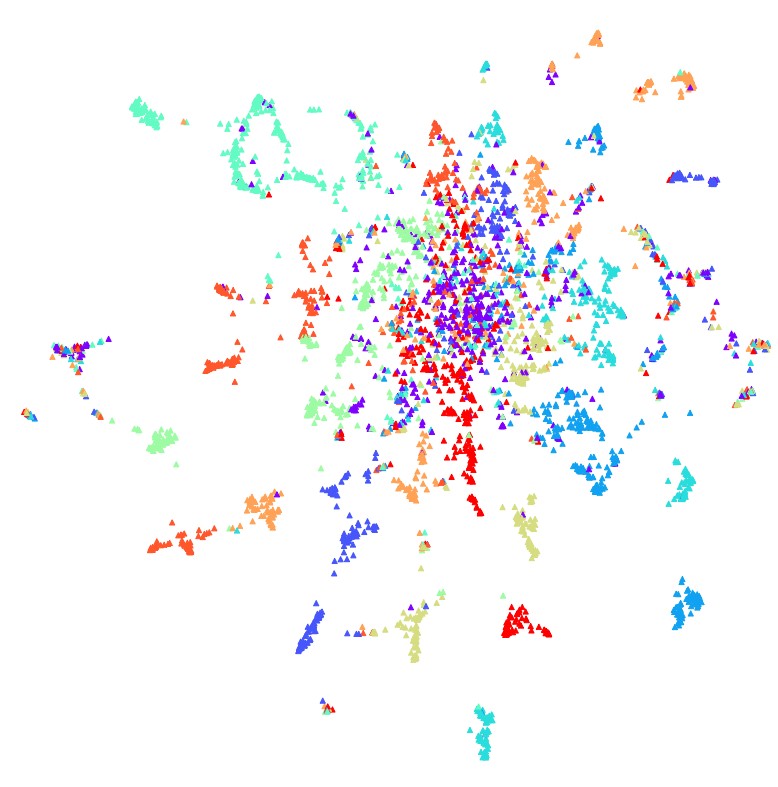

Figure 7: Rice31 ground truth class labels.

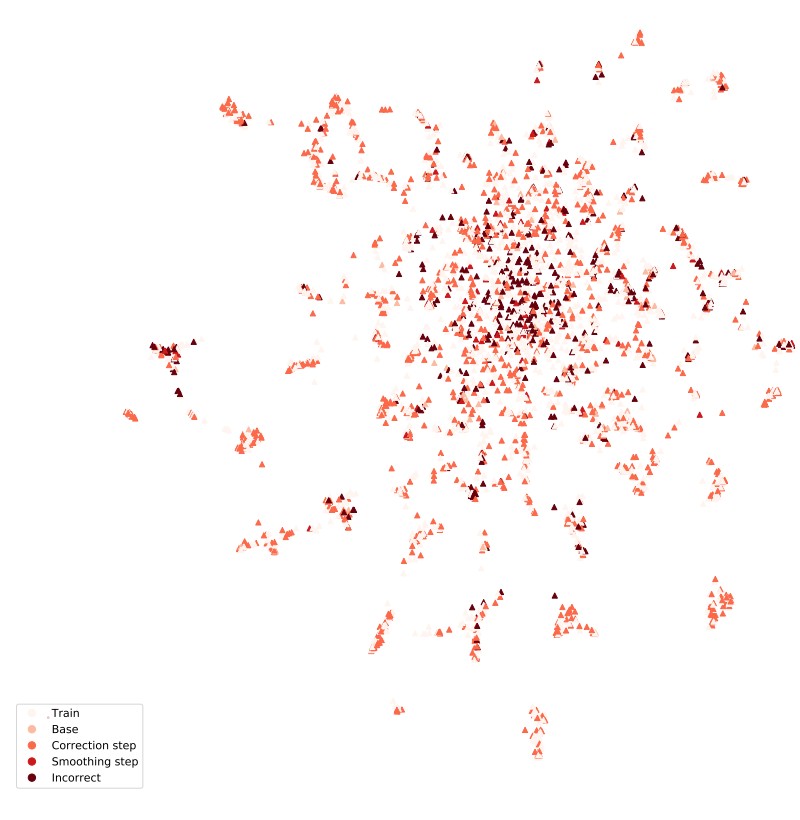

Figure 8: Linear-SE + C&S prediction performance on Rice31.

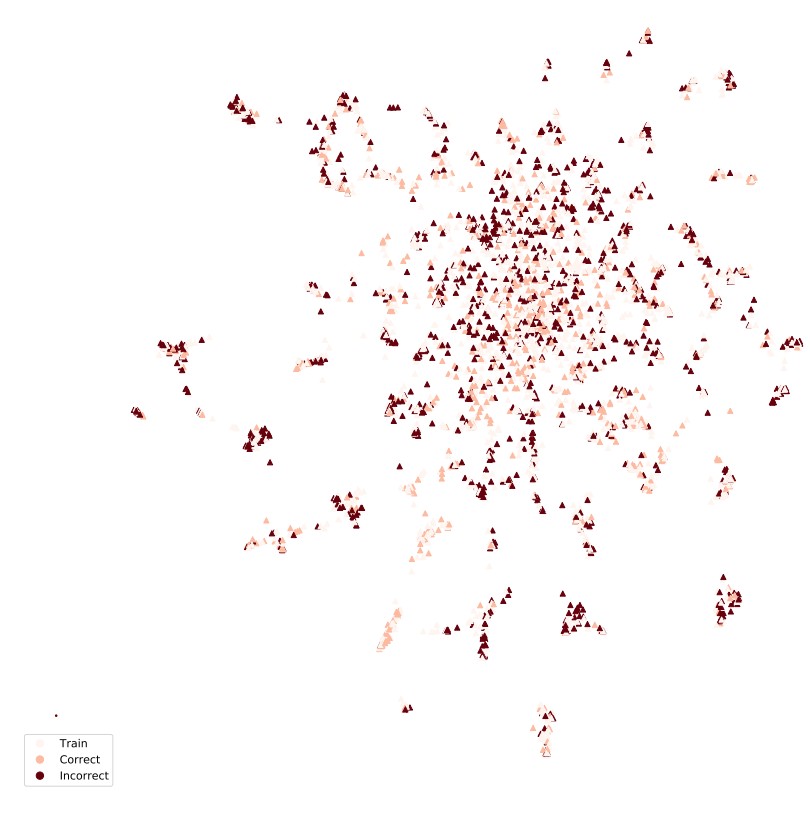

Figure 9: GCN-SE prediction performance on Rice31.

