# OpenReview forum: "Combining Label Propagation and Simple Models out-performs Graph Neural Networks"
_ICLR.cc/2021/Conference — ICLR 2021 Poster_

### Official Review · AnonReviewer2 · 2020-10-16
**Interesting results, but experiments do not support claims.**

**Rating:** 7
**Confidence:** 4

**Review:**

## Summary
This paper presents C&S method that predicts node labels in the transductive semi-supervised node classification setting. C&S uses the three-stage-pipeline approach. First, label probabilities are predicted with simple and scalable classifiers such as MLP. Then, the predicted errors are diffused over graphs. Finally, the labels are further smoothened to give the final node label prediction. The authors demonstrate that their simple C&S approach beats many existing GNN approaches.

Although the paper makes interesting claims, I also have many concerns about whether the experiments support the central claims. The experiments need to be well-organized and improved to support the claims.

## Pros:
1. Provides evidence that diffusing label information is useful and can even beat conventional GNNs.
2. The results are promising, especially on the OGB products dataset.
3. Let the community rethink the value of the classical approach of the label diffusion for semi-supervised node classification.

## Concerns
1. For many datasets, C&S with the plain linear base predictor does not give good performances, This implies that spectral embeddings capturing graph structure are important for C&S. However, in the abstract and introduction, I got the impression that the base predictor ignoring graph structure is enough for C&S to achieve good performance.
2. If spectral embeddings benefit C&S, why not use the embeddings also for the other GNN models? The central claim of the paper is that C&S algorithm (label diffusion) could be better than GNNs (feature diffusion), but there is a confounding factor that GNNs are not utilizing spectral embeddings as model input.
3. In Section 2.4, the authors mentioned 15 seconds for 1M edges, but in Table 1, on the OGB Arxiv dataset (with about 1M edges), time is only 9.89s. Why is that?
4. Computing spectral embeddings is quite expensive on large graphs, while the authors claim it is fast. To demonstrate this, in Table 1, please make sure to include the time for computing spectral embeddings (and all other computations in C&S), and compare the time with existing GNN models.
5. Table 1 is confusing because C&S is using validation labels, and the absolute numbers are not reported. Reporting absolute numbers, time, and SOTA method names is important because SOTA will quickly change over time.
6. The model used for the papers100M dataset is not mentioned. What model did you use to achieve 65.33% accuracy? Why don't the authors treat papers100M dataset in the same way as the other datasets (by including the result in Tables 1 and 2)?
7. Ablation studies of doing only C or S is missing. Table 3 is kind of doing ablation studies of only retaining the S part, but I’d like to see more pure ablation studies. How much does each component contribute to the final accuracy?
8. Sensitivity of scaling hyper-parameters $s$ in FDiff-scale and the number of iterations in graph diffusion is not mentioned. Especially, $s$ seems to be quite important.
9. In Section 3.2, the authors claim that GNNs do not take advantage of validation labels; however, this is not immediately obvious to me. Please justify this through experiments.
10. In Table 3, what if you use spectral embeddings for Plain GCN?
11. In Section 1.1, “but these methods focus on settings of low label rates” is unclear, and seems tangential to the actual methods.
12. “Methods” column in Table 2 is unclear to me.
13. In Table 2, why did the authors mark 88.73 in bold (89.05 seems to be the best)? If Auto-scale is consistently worse than FDiff-scale, does it make sense to drop it entirely?
14. Figure 3 is hard to understand.
15. Standard deviations are not included. There should be randomness coming from MLP training.

====

Post-rebuttal: I am happy to accept this paper after seeing my concerns are mostly addressed.

---

> ### Author Response · Authors · 2020-11-21
> **Response to Reviewer 2 (1/2)**
>
> We thank the reviewer for the many detailed comments. We have addressed them and the paper has improved substantially.
>
> The reviewer wrote that our experiments do not support the claims. We do not believe this to be true, and we detail the reasons below. The 15 concerns raised by the reviewer are largely minor clarification issues. Addressing them will make the paper better, but we do not find any of them to be fundamental issues. More details are below.
>
> A. Spectral embeddings (points 1, 2, 4, and 10)
>
> We thank the reviewer for pointing out the role of spectral embeddings, which we should have made more prominent and clear in the paper. We use spectral embeddings to demonstrate an additional way to leverage traditional methods. They are relatively cheap to compute, help us achieve state-of-the-art performance, and are largely unused in most graph learning algorithms.
>
> (Point 1)
> > For many datasets, C&S with the plain linear base predictor does not give good performances, This implies that spectral embeddings capturing graph structure are important for C&S. However, in the abstract and introduction, I got the impression that the base predictor ignoring graph structure is enough for C&S to achieve good performance.
>
>
> To be clear, we do use the graph structure during the C&S part of our framework (through performing diffusions). What we don’t do is use the graph structure during supervised feature learning. This is what makes most GNN models expensive to train and our models cheap. We have clarified this in the text.
>
> In addition, we emphasize that we can still achieve strong results on many datasets without spectral embeddings. In the paper, this is the “Plain Linear” model. In addition, we also ran experiments with “Plain MLP” (i.e., MLP without spectral embeddings). A full table will be provided in the Appendix as Table 7 of our updated paper, and here is a summary of the results:
>
> The plain MLP model obtains better results than the standard GCN (even with the spectral embeddings from A.2) in 7/8 datasets (other than arxiv), and outperforms SOTA results on Products, Cora, Rice31, and USCounty. Email is not included in these comparisons as it has no base features.
>
> Once again, we’d like to highlight the ogbn-products results, which drop from 84.18 with spectral embeddings to 83.8 without. This is still 1.2% higher than the previous SOTA, with an order of magnitude less parameters and runtime.
>
>
> (Points 2 and 10) Regarding the use of spectral embeddings in conjunction with other methods.
>
> One could use these embeddings with other methods; however, this will not make them any faster to train, which is a major problem that we address. We note that when possible, we have attempted augmenting the SOTA GNNs with spectral embeddings; for instance, the “SOTA” entries for Email and Rice31 both involve GNNs augmented with spectral embeddings. We have also added additional results for “Plain GCN” augmented with spectral embeddings in the Appendix. In general, this does not improve GCN performance much.
>
> Also, as mentioned above, we do not rely on spectral embeddings to get good performance. For this reason, we have focused on how people typically use these methods, which does not involve augmenting the base features. One of our main claims is that basic GNN level performance is achievable with a much simpler/interpretable method, and this is backed up by our experiments.
>
> Finally, there is a common misconception that spectral embeddings are often expensive to compute. While they are expensive to compute with naive algorithms, there are a number of robust numerical algorithms (namely, Krylov subspace methods) that can approximately compute leading eigenvectors with only (sparse) matrix-vector products. In practice, we use Krylov methods with a tolerance of 1e-5, and this converges in a constant number of iterations (fewer than 100).
>
> B. Ablation of C&S (point 7)
>
> We will update this in the appendix (table 6). In general, both the “Correct” and “Smooth” stages are important for getting high performance.

---

> ### Author Response · Authors · 2020-11-21
> **Response to Reviewer 2 (2/2)**
>
> C. Various small clarifications (all other points)
>
> (Points 3, 4, and 5) On Table 1.
>
> There are some misunderstandings here, and we have tried to make everything clear. The point of Table 1 is to give a sense of the datasets and types of gains that are possible with a simple method. The exact numbers are provided in Tables 2 and 4. We claimed that we can often out-perform (in terms of both accuracy and training time) SOTA methods with simpler techniques, and this is backed up by Table 1.
>
> We will add specific runtimes for GNN models in the Appendix. However, this is not always possible for all datasets/architectures, as some SOTA models use unorthodox frameworks.
>
> Some more direct clarifications are:
> -- Table 1 shows results using the validation set.
> -- Runtimes are those that are needed for the model chosen. Some of these use spectral embeddings and some do not.
> -- The SOTA models are described at the beginning of Section 3.
> -- Absolute numbers are reported in Table 2 for SOTA and the appendix for our method.
> runtimes are those that are needed for the model that achieves that performance.
>
>
>
> (Point 6) Clarity of Papers100M experiments
> We used an MLP model with spectral embeddings, combined with C&S (FDiff-scale variant). We didn’t include the results in the other tables as unlike the other datasets, as we were unable to run an exhaustive set of experiments due to computational constraints.
>
> (Point 8) On hyperparameters
> We select the hyperparameters based on hyperparameter search over validation set. The number of iterations is fixed to be 50. We will update this in the beginning of Section 3.1.
>
> (Point 11) On the use of validation labels.
> At inference, our C&S model can directly use validation labels as inputs, with the hyperparameters selected during training time over validation sets. However, this is not the case for GNNs, as there is no obvious way to add validation labels as inputs only during inference time to further improve performance.
>
> (Points 12 and 13) On Table 2.
> We interpret each model as a plain neural network plus a certain modification, i.e., the “method.” For example, “FDiff-scale” with “Plain Linear” corresponds to the plain linear base prediction (following Section 2.1) with the FDiff-scale post-processing (following Section 2.2).
> There was a typo in the bolding, and thanks for catching it. Although FDiff-scale performs best on the datasets where we achieve SOTA, autoscale still outperforms FDiff-scale on datasets like ogbn-arxiv or Citeseer.
>
> (Point 14) On Figure 3.
> Figure 3 visualizes which nodes are corrected by the C&S steps. The left panel shows the ground truth classification results, with the three boxed regions zoomed in to be the right panels. The different colors in the small panels correspond to at which stage the node becomes classified as correct (base, C or S) or never corrected (wrong), as well as the train nodes. Please see Section 3.3 for a more detailed description.
>
> (Point 15) Variance in performance
> We were limited by space, but the standard deviations are typically very small on datasets without random splits (e.g., <0.12 on ogbn-arxiv with C&S, and <0.07 on ogbn-products). On datasets with random splits our improvements are almost always still statistically significant.

---

> ### Comment · AnonReviewer2 · 2020-11-21
> **Thanks for addressing my concerns.**
>
> Thank you for your response. My concerns are mostly addressed, and I am happy to increase my score.  I hope the authors reflect all the promised points in the final version of the paper. One minor point below:
>
> Re: Finally, there is a common misconception that spectral embeddings are often expensive to compute. While they are expensive to compute with naive algorithms, there are a number of robust numerical algorithms (namely, Krylov subspace methods) that can approximately compute leading eigenvectors with only (sparse) matrix-vector products. In practice, we use Krylov methods with a tolerance of 1e-5, and this converges in a constant number of iterations (fewer than 100).
>
> I hope the authors can give the actual computation time for each dataset to strengthen the claim.

---

> > ### Author Response · Authors · 2020-11-21
> > **Thanks for reading the response :)**
> >
> > > I hope the authors can give the actual computation time for each dataset to strengthen the claim.
> >
> > We will add these numbers soon.
> >
> > We'd like to add one more clarification about the role spectral embeddings play in our paper. The reviewer made a very good point that we should be including runtime of spectral embedding (we were inconsistent about whether we did in Table 1). And indeed, spectral embeddings make up a significant portion of the runtime compared to training the linear layer or MLP.
> >
> > However, there are several aspects of our method we would like to emphasize. 2 of them, as you've pointed out, are runtime and performance. However, there are other advantageous aspects of our model as well, such as simplicity, interpretability, and parameters.
> >
> > Thus, although spectral embeddings may make the runtime worse (for a modest increase in performance on some datasets), it does not reduce the simplicity nor the interpretability of the model.
> >
> > If runtime is the primary concern, there are alternative choices of embeddings (that we did not discuss in the paper) that incur negligible computational cost. For example, a simple diffusion embedding (e.g: SGC) matches (and sometimes exceeds) spectral embeddings results on some datasets when combined with our framework (like ogbn-arxiv, where a MLP+diffusion+C&S gets to 72.7).
> >
> > Finally, the reviewer (as well as reviewer 4) provided valuable feedback about how several of the tables/naming conventions used in this paper are confusing. We agree, and beyond the changes in the revised version, are thinking of how to restructure the data for improved clarity in the final version.
> >
> > We'd like to thank the reviewer again for the detailed reading and comments on our paper.

---

### Official Review · AnonReviewer4 · 2020-10-26
**Official Blind Review #4**

**Rating:** 7
**Confidence:** 4

**Review:**

Summary
-------------
A method to perform transductive node classification in graphs is presented. This is the case in which a model is trained on a large graph in which only some nodes are labeled, and the inference task is to label the remaining nodes correctly.
The method uses a simple per-node prediction model $f$ (ignoring the graph structure) and then two post-processing steps: (1) propagating information about errors made by $f$ on labeled nodes to its neighbours, and (2) "smoothing" predictions, e.g., encouraging that adjacent nodes have the same label.
In experimental results on a variety of datasets, the method has comparable to or better performance than substantially more complex Graph Neural Networks, while requiring 1-2 orders of magnitude fewer parameters and less compute time.

Strong/Weak Points
-------------
* (+) The method is simple and largely well-described, and it is interesting and notable that it is extremely competitive in this setting.
* (+) The experiments range over a substantial set of datasets, models and ablations, providing ample evidence that this is not just a fluke.
* (-) The zoo of model variants is not well-described and confusingly named ("Base Model" <-> "Base Prediction", "Linear" vs. "Plain Linear", etc.). This part of the paper could be substantially improved by better naming and editing.
* (-) The title (mainly, + some of the abstract, and a bit of the text) is overly focused on the relation to GNNs instead of creating clarity about the method. This is somewhat misplaced as it is not applicable to a majority of GNN use cases (such as graph-level uses for property predictions of molecules, proteins, etc.; or the modelling of mathematical proofs or programs; or anything that isn't transductive node classification). The paper is interesting in itself, but puts focus on its weaknesses by this aggressive sales pitch.

Recommendation
-------------
I think this paper should be accepted for publication, as it serves as a much-needed reality check for some of the more frothy publication trends in GNN research at the moment.
However, I would strongly recommend to reduce the sales focus. For example, "Label Propagation is all you need for Transductive Node Classification" would be a what-counts-as-witty-in-the-ML-community title that is more informative and less combative than the current one.

Questions
-------------
* Sect 3: I believe that "plain linear" refers to an initial per-node prediction layer that operates only on the raw node features, distinguished from the "linear" variant which uses the raw node features and additionally the spectral features as described in Sect. 2.4. Is this correct? It would be helpful to update the text to clarify this.
* Sect 3: What does "Plain GCN" refer to? How is it different from just writing "GCN"?
* Sect 3: Does "Base Prediction" refer to the case of not using any label propagation?

Detail Feedback
-------------
* Sect 2: For reproducibility purposes, it would be helpful to specify values for $\mu$, the number of iteration steps.
* Table 2: This would be much easier to read with either all results next to each other (resizebox may help here) or with a spatial separation between the two sets of results.

---

> ### Author Response · Authors · 2020-11-21
> **Response to Reviewer 4**
>
> We thank Reviewer 4 for the positive comments, both on the extensive evaluation as well its overall place in the GNN research literature.
>
> We agree that the current title does not make clear some of the relatively limited settings in which our model applies. We have made a sincere effort in the abstract and rest of the text to discuss the limitations. We think a new title of “Combining Label Propagation and Simple Models for Fast and Accurate Node Classification” would avoid unnecessary combativeness and would appreciate feedback on this choice.
>
>
>
> > Sect 3: I believe that "plain linear" refers to an initial per-node prediction layer that operates only on the raw node features
>
> Correct. We have updated the text to clarify.
>
> > Sect 3: What does "Plain GCN" refer to? How is it different from just writing "GCN"?
>
> “Plain GCN” refers to a variant that doesn’t make use of any additional input embeddings but still makes use of tricks like residual connections and dropout that improve practical performance. It serves as a fairly strong baseline of existing out-of-the-box GNN capabilities. We have updated the text to clarify.
>
> > Sect 3: Does "Base Prediction" refer to the case of not using any label propagation?
>
> Yes, that is the performance of the model before we apply any label propagation steps.
>
> > Sect 2: For reproducibility purposes, it would be helpful to specify values for \mu, the number of iteration steps.
>
> We use 50 iterations for all of our experiments (except for papers100M, for computational reasons), and we have specified this in the text. The label propagation variants we use are empirically known to converge quickly and reliably, so we did not tune it as a hyperparameter.

---

### Official Review · AnonReviewer1 · 2020-10-27
**Review Comment #1**

**Rating:** 6
**Confidence:** 3

**Review:**

(Added on 11/29/2020)

**Post Rebuttal Comment**

I thank the authors for sincerely replying to my review comments. The authors' answers were reasonable to me.

-------------------------------------

**Review Summary**

This paper has conducted an extensive comparison with many existing GNNs with a wide variety of datasets. It supported an underlying hypothesis that using label information directly is helpful for better performance. Several studies have questioned the architecture of the current standard GNNs. Considering that, it is important to rethink whether the modules taken for granted are redundant. This research has given empirical evidence that an end-to-end node aggregation scheme might not be a critical component for good performance. It deepens the understanding of GNN models.

**Summary and Contributions**

It proposed Correct and Smooth (C&S) for solving graph-learning problems. The model consists of a shallow model that ignores underlying graph structures followed by two label correction based on label propagation algorithms. This paper tested C&S with several benchmark datasets and compared it with popular GNNs. Also, this paper applied the proposed correction methods to GNNs and showed that it provided performance gain.

**Claim**

If I understand correctly, the main claims of this paper are as follows. I will evaluate this paper whether theories and experiments support these claims.

- Claim 1: C&S performs comparably in classification tasks with existing models.
- Claim 2: C&S is computationally cheap compared to existing models.
- Claim 3: C&S improves the predictive performance of existing models.

**Soundness of the claims**

Can theory support the claim?
- This paper did not provide a theoretical justification for the proposed method.

Can empirical evaluation support the claim?
- Claim 1, 2 are supported by experiments in Table 2 (Section 3.2) and Table 1. Remarkably, this paper used nine datasets, which have a relatively wider variety compared to standard GNN papers. I agree that C&S + Autoscale or FDiff-scale can achieve comparable performance with a small number of parameters.
- Claim 3 is supported by experiments in Table 5 (Section 3.3). As pointed out by authors, the performance gain obtained by incorporating C&S to GAT and GCNII is modest. The difference in performance between GCNII and GCNII+C&S is within the standard deviation. So, I do not think the experimental results well support claim 3.

**Significance and novelty**

Relation to previous work

- This paper mentioned the following models as related works.
  - label propagation: APPNP (Klicpera et al, 2018) (Section 1.1, Paragraph 1), GCN-LPA (Wang & Lescovec, 2020)), C-GCN (Jia & Benson, 2020) (Section 1, Paragraph 4)
  - Markov random field: Qu et al. (2019), Gao et al. (2019) (Section 1, Paragraph 4)
  - Shi et al. (2020) (Section 1, Paragraph 4)
- As pointed out by this paper, several existing methods have incorporated the label propagation algorithms to GNNs, such as APPNP, GCN-LPA, and C-GCN. The authors discussed the difference between these methods in Section 1, Paragraph 4, and claimed that the proposed method is advantageous in computational cost. The author tested in Fig.2. I want to confirm which methods are used as the OGB Leaderboard in Figure 2.
- Regarding the difference from C-GCN, this paper discussed that while Jia & Benson (2020) focused on regression tasks, this study is interested in classification tasks.
- Iscen et al. (2019) employed the idea of combining label propagation and MLP in semi-supervised learning. However, the problem setting of Iscen et al. (2019) is different from this paper in that the former one does not have underlying graph structures a priori while the latter one has.
[Iscen et al., 2019] A. Iscen, G. Tolias, Y. Avrithis, O. Chum. "Label Propagation for Deep Semi-supervised Learning," CVPR 2019

Relevance to community
- Several studies have questioned the architecture of the current standard GNNs. For example, SGC (Wu et al., 2018) questioned the role of non-linearity, gfNN (NT and Maehara, 2019) questioned the interleaving of node aggregations and non-linear transformations. If I understand correctly, this study has a similar spirit in common.

**Correctness and Clarity**

Is the theorem correct?
- No theorems are provided.

Is the experimental evaluation correct?
- In the paragraph starting with "Under a Gaussian assumption ..." in Section 2.2, the authors hypothesized why plain C&S is insufficient for classification tasks from the difference between regression and classification tasks. Based on this discussion, they introduced Autoscale and FDiff-scale. Table 2 showed that the C&S methods do not perform comparably with existing methods without the autoscale or FDiff-scale methods in classification tasks. If the above hypothesis is correct, I expect that the plain C&S performs well in regression tasks. However, we cannot confirm the hypothesis because no experiments are performed on regression tasks.

Is the experiment reproducible?
- Although the experiment code is not provided, the experiments' details are provided, such as hyperparameters and train/validation/test splits.

**Clarity**

Can I understand the main point of the paper easily?
- Yes

Is the organization of the paper well?
- Yes

Are figures and tables appropriately made?
- Yes

**Additional feedback**

Section 3 Paragraph (Base predictors and other models) Line 7: For For Cora ... → For Cora ...

---

> ### Author Response · Authors · 2020-11-20
> **Response to Reviewer 1**
>
> We thank Reviewer 1 for the detailed summarization and checks. Based on the review, we only see a concern in that the third claim on improving the performance of existing models is not well-supported.
>
> We emphasize that the third claim is not the main result in our paper. What we really want to say is that our model is flexible enough to be used with existing models. This is backed up by the experiments in Section 3.3 and Table 5. The modesty of the gains is meant to highlight that we do not believe that big GNN models are capturing much structure beyond what is in our model. We will update the paper to reflect this messaging.
>
> In addition, for US-county, we'd like to emphasize that the vast majority of the variance comes from random splits. In addition, as C&S is applied as a postprocessing step on top of the the existing model, we can also examine the standard deviation of the improvement from the base model. As an example (not the actual numbers), if the GCNII results are [90.0, 110.0] and the C&S results are [100.0, 110.0], then the std as reported in the paper would be 10.0, but the std of the improvement would be 0.
>
> Calculating this metric, it is far lower, at <0.2.
>
> > I want to confirm which methods are used as the OGB Leaderboard in Figure 2.
>
> The SOTA results (at the time of submission) were UniMP_Large for ogbn-arxiv and UniMP for ogbn-products

---

> > ### Comment · AnonReviewer1 · 2020-11-21
> > **Response to authors' comments.**
> >
> > I thank the authors for considering my review comments seriously. Also, I am sorry that I should have to clarify my concerns, e.g., by listing them up. My concerns were mainly two folds:
> >
> > - (1) Claim 3 is not well-supported by Table 5
> > - (2) the explanation of the paper why the vanilla C&S (i.e., without Autoscale nor FDiff-scale) does not work is practically correct.
> >
> > The authors' comment has kindly answered (1).
> >
> > Regarding (2), what I thought was that the following sentence might not be practically justified because they did not apply the vanilla C&S in regression tasks.
> >
> > > This type of propagation is provably the right approach under a Gaussian assumption in regression problems (Jia & Benson, 2020);
> >
> > However, I am also wondering that I should ask this paper to conduct experiments on regression tasks because this paper's focus is classification tasks.

---

> > > ### Author Response · Authors · 2020-11-21
> > > **Further response**
> > >
> > > If I understand the reviewer's concerns correctly, it is primarily about further understanding of Autoscale or FDiff-scale, and when they are needed. This is a good point, and there are 2 facets to this.
> > >
> > > ------
> > >
> > > 1. > This type of propagation is provably the right approach under a Gaussian assumption in regression problems (Jia & Benson, 2020);
> > >
> > > This sentence refers to a previous paper (Residual Correlation in Graph Neural Network Regression), where they assume on regression tasks that the error is distributed as a multivariate gaussian, and demonstrate that their residual correlation (a Laplacian diffusion without any kind of scaling) is the right approach.
> > >
> > > However, this is obviously a very strong assumption, and it's also not clear what a similar gaussian assumption would look like for classification tasks.
> > >
> > > ----
> > >
> > > 2. The second aspect is that the paper does not include empirical experiments showing that a similar residual correlation to (Jia & Benson, 2020) does not suffice. Table 2 does not actually compare a C&S variant without Autoscale/FDiffScale - "base prediction" is the prediction of the model before we apply each part of C&S. This is a good point and we will include further ablations in the Appendix.
> > >
> > > The following is an ablation of a few of the settings. We refer to a  variant using a similar residual correlation to (Jia & Benson) as "no scaling" in the below. As you can see, although that variant still improves performance, the performance is dramatically worse than variants using Autoscale or FDiff-Scale.
> > >
> > > ---
> > >
> > > Arxiv, Plain Linear (C&S no scaling):  52.32 (Base Accuracy) -> 57.9 (Accuracy after "Correct" step) -> 69.52 (Accuracy after "Smooth" step)
> > >
> > > Arxiv, Plain Linear (C&S Autoscale): 52.32 (Base Accuracy) -> 66.89 (Accuracy after "Correct" step) -> 71.11 (Accuracy after "Smooth" step).
> > >
> > > ---
> > >
> > > Arxiv, MLP (C&S no scaling): 71.51 (Base Accuracy) -> 71.61 (Accuracy after "Correct" step) -> 72.13 (Accuracy after "Smooth" step)
> > >
> > > Arxiv, MLP (C&S Autoscale): 71.51 (Base Accuracy) -> 71.97 (Accuracy after "Correct" step) -> 72.62 (Accuracy after "Smooth" step)
> > >
> > > ---
> > > Products, Plain Linear (C&S no scaling ): 47.73 (Base Accuracy) -> 50.44 (Accuracy after "Correct" step) -> 72.32 (Accuracy after "Smooth" step)
> > >
> > > Products, Plain Linear (C&S FDiff-scale): 47.73 (Base Accuracy) -> 80.97 (Accuracy after "Correct" step) -> 82.54 (Accuracy after "Smooth" step)
> > >
> > > ---
> > >
> > > Running C&S on regression tasks is an interesting idea - if the error is not Gaussian, then Autoscale/FDiff-scale may still provide some gains. We will consider doing so for the final version of the paper.

---

> > > > ### Comment · AnonReviewer1 · 2020-11-22
> > > > **Comment on the second question**
> > > >
> > > > I thank the authors for prompt reply and sharing the result of the ablation study. I understand that the sentence intended to refer to the previous study (Jia & Benson, 2020). I also understand that the previous study is a side evidence that the correction part (eq.(2)) may work for regression tasks, although the previous study and this study (especially eq.(2)) differ in how to apply the label propagation algorithm.

---

> > > > > ### Author Response · Authors · 2020-11-22
> > > > > **Thank you for the clarification**
> > > > >
> > > > > We agree that C&S may work for regression tasks - we will consider adding another task for the final version of the paper.
> > > > >
> > > > > From our reading of the comments, it seems that the reviewer's concerns have been clarified. We thank the reviewer for the positive comments on how our paper deepens the understanding of GNN models, our convincing and extensive experimental evidence across a wide range of datasets, and clarity of our paper.
> > > > >
> > > > > Please let us know if there are any remaining concerns, experimental results, or clarifications that we could make to strengthen our paper.

---

### Official Review · AnonReviewer3 · 2020-11-02
**an interesting work, but needs deeper study**

**Rating:** 6
**Confidence:** 4

**Review:**

This paper shows modified label propagation can perform better than GCN. The idea is as follows: it first uses MLP on node features to get the initial labels, and then use two steps--correction and smoothness to postprocessing the labels. And this postprocessing is based on the traditional label propagation algorithm. It shows that this simple method matches GCN performances on various datasets.

1) To my understanding, this method can be viewed as proposing a 'better' initialization for label propagation.  In label propagation, we usually start from random for these unlabeled nodes, while this method use MLP over node features to get the initial guesses for these unlabeled node's labels. Therefore in terms of novelty I am not quite about the contribution.

2) Since this work is on improving label propagation for node classification, various label propagation algorithms are better to compared with to show the benefit of MLP other than just comparing with GCN methods.

3) Can the proposed method be used for inductive setting? or other graph tasks other than node classification? The power of GCN is that it learns node embeddings and then used for various end-tasks. Also does the features for each node need to be very good to achieve good performance? If that is case, what if we just use features with some traditional ML algorithms such as kernel SVM? How about the results on some social network where the graph has strong connections between nodes? Without answering all these questions, it is hard to tell the usability of the proposed method, and get readers a deep understanding why GCN is not performing well with this simple heuristics.

---

> ### Author Response · Authors · 2020-11-21
> **Response to Reviewer 3**
>
> Thank you for the comments. We have some clarifications below.
>
> > To my understanding, this method can be viewed as proposing a 'better' initialization for label propagation.
>
> We agree that the “Smooth” part can be viewed as a better initialization for label propagation, and it is remarkable that this, by itself, already improves performance significantly. Furthermore, we disagree that the “Correct” part can be viewed as a “better initialization” for label propagation - there is no analogue of a “residual error”  in traditional label propagation approaches, as there is no initial prediction. This is a fundamentally different form of label propagation than what is commonly done.
>
>
> > Since this work is on improving label propagation for node classification, various label propagation algorithms are better to compared with to show the benefit of MLP other than just comparing with GCN methods.
>
> Our work provides a bridge between feature learning and label propagation, so we don’t restrict ourselves in our comparisons. Thus, we compare to the state of the art on standard benchmarks, such as the Open Graph Benchmark and the Cora/Pubmed/Citeseer datasets. On these benchmarks, the state of the art methods are GNN-based architectures and thus, we primarily compare ourselves to those baselines.
>
> We do compare against a standard label propagation algorithm that is also a part of our methods. Other label propagation algorithms have similar performance generally. If there is a specific method that the reviewer would like to see in a comparison, please let us know.
>
> > Can the proposed method be used for inductive setting? or other graph tasks other than node classification?
>
> The focus in this paper is on transductive node classification benchmarks with homophilous structure. Although this is a specific class of benchmark, this is also the most common class of tasks. For example, four out of the five OGB node classification tasks fit this description, as do the Cora, Pubmed, and Citeseer datasets.
>
> That being said, our methods can be used in some inductive settings. For example, if a graph evolves over time (a common inductive setting), one could easily use the same base classifier, and there is little overhead to running the label propagation steps if new or additional labels become available.
>
> One of our points is that these types of simple methods can outperform graph neural networks. Adapting our approach for other tasks is an opportunity for future research.
>
>
>
> > Also does the features for each node need to be very good to achieve good performance? If that is case, what if we just use features with some traditional ML algorithms such as kernel SVM?
>
> We have reported results using logistic regression as the base classifier (Plain Linear and Linear in table 2), which is a traditional ML algorithm. One could also use kernel SVMs, and this flexibility is an advantage of our approach. We have already shown that existing broadly-used classifiers are sufficient for good performance, but other ones could certainly be used as well.
>
> > How about the results on some social network where the graph has strong connections between nodes?
>
> We evaluate on two social network datasets where the edges represent strong connections; these are the email and Rice datasets. All of our datasets have strong homophilous connections (i.e., have strong connections between nodes), and many of them are common and standard benchmarks.

---

### Author Response · Authors · 2020-11-21
**Overall Response to Reviewers**

We thank all the reviewers for the valuable feedback. Overall, we find that the reviewers agree on the interesting nature of our central claim that simple models, combined with label propagation, can outperform conventional graph neural networks for node classification tasks at substantially less cost. Moreover, Reviewers 1, 2, and 4 noted that this paper will potentially be influential to the field. Reviewers 1 and 4 also point out that this central claim is supported by extensive experiments over a large range of datasets.

The critical feedback from the reviewers identified a number of points in the paper that could be clarified or backed up with additional experimental details. Review 2 also suggested several further ablation experiments to solidify the claims. Addressing these issues have substantially improved the paper, and our fundamental results remain the same. Below, we respond to individual concerns with clarifications and additional results, which will be added to the paper shortly.

---

### Decision · Program_Chairs · 2021-01-07
**Final Decision**

**Decision:**

Accept (Poster)

**Comment:**

After rebuttal the reviewers unanimously agree that this is a strong paper and should be accepted